# A New Plasmacytoid Dendritic Cell-Based Vaccine in Combination with Anti-PD-1 Expands the Tumor-Specific CD8+ T Cells of Lung Cancer Patients

**DOI:** 10.3390/ijms24031897

**Published:** 2023-01-18

**Authors:** Dalil Hannani, Estelle Leplus, David Laurin, Benjamin Caulier, Caroline Aspord, Natacha Madelon, Ekaterina Bourova-Flin, Christian Brambilla, Elisabeth Brambilla, Anne-Claire Toffart, Karine Laulagnier, Laurence Chaperot, Joël Plumas

**Affiliations:** 1PDC*line Pharma, 38000 Grenoble, France; 2Recherche et Développement, EFS, 38000 Grenoble, France; 3Institute for Advanced Biosciences, Université Grenoble-Alpes, INSERM U1209, CNRS UMR 5309, 38000 Grenoble, France; 4Groupe EpiMed, Université Grenoble-Alpes, INSERM U1209, CNRS UMR 5309, 38000 Grenoble, France; 5Centre Hospitalo-Universitaire Grenoble-Alpes, Université Grenoble-Alpes, 38000 Grenoble, France

**Keywords:** therapeutic cancer vaccine, lung cancer, plasmacytoid dendritic cells, immunotherapy, checkpoint inhibitors

## Abstract

The purpose of immune checkpoint inhibitor (ICI)-based therapies is to help the patient’s immune system to combat tumors by restoring the immune response mediated by CD8+ cytotoxic T cells. Despite impressive clinical responses, most patients do not respond to ICIs. Therapeutic vaccines with autologous professional antigen-presenting cells, including dendritic cells, do not show yet significant clinical benefit. To improve these approaches, we have developed a new therapeutic vaccine based on an allogeneic plasmacytoid dendritic cell line (PDC*line), which efficiently activates the CD8+ T-cell response in the context of melanoma. The goal of the study is to demonstrate the potential of this platform to activate circulating tumor-specific CD8+ T cells in patients with lung cancer, specifically non-small-cell lung cancer (NSCLC). PDC*line cells loaded with peptides derived from tumor antigens are used to stimulate the peripheral blood mononuclear cells of NSCLC patients. Very interestingly, we demonstrate an efficient activation of specific T cells for at least two tumor antigens in 69% of patients irrespective of tumor antigen mRNA overexpression and NSCLC subtype. We also show, for the first time, that the antitumor CD8+ T-cell expansion is considerably improved by clinical-grade anti-PD-1 antibodies. Using PDC*line cells as an antigen presentation platform, we show that circulating antitumor CD8+ T cells from lung cancer patients can be activated, and we demonstrate the synergistic effect of anti-PD-1 on this expansion. These results are encouraging for the development of a PDC*line-based vaccine in NSCLC patients, especially in combination with ICIs.

## 1. Introduction

Immunotherapies with immune checkpoint inhibitors (ICIs), such as antibodies against the programmed cell death protein-1 (anti-PD-1) or its ligand PD-L1, have changed the therapeutic landscape of cancer. This is notably the case in lung cancer, which is the leading cause of cancer-related mortality worldwide, with an estimated 1.8 million cancer deaths in 2020 [1]. Unresectable metastatic non-small-cell lung cancer (NSCLC) patients without genetic alterations are now treated, in first-line treatment, with ICIs in monotherapy or in combination with chemotherapy depending on the level of tumor PD-L1 expression [2,3]. Despite the observed clinical benefits, many patients still do not respond, develop resistance, and progress.

One promising way to increase the number of responder patients and prolong their survival is to stimulate the patient’s immune system with cancer vaccine approaches. Indeed, as strongly suggested by the treatment of patients with ICIs in a neo-adjuvant setting, the unleashing of the patient’s own immune system is responsible for tumor eradication [4,5].

The main cancer vaccine approach consists of priming and activating the patient’s tumor-specific cytotoxic CD8+ T lymphocytes with dendritic cells (DCs) since they are professional antigen-presenting cells [6]. Interestingly, boosting antitumor immunity could benefit all patients regardless of the PD-L1 expression by tumor cells and tumor-infiltrating leukocytes. However, cancer vaccines based on autologous DCs have proven difficult to produce and have shown little clinical benefit [7]. Allogeneic DC-based vaccines are proposed as alternatives to their autologous counterparts because they overcome several issues, including the lack of reproducibility and the difficulty of production from patients’ blood [7]. In addition, the allogeneic response due to the expression of mismatched HLA on allogeneic DCs is expected to generate an activation stimulus that strengthens the stimulation of antigen-specific CD8+ T cells (ASTCs).

We have been developing a new allogeneic DC-based cancer vaccine, which has shown a strong capacity to activate rare tumor ASTCs. This vaccine consists of a cell line, named “PDC*line”, originating from the blood of a patient with plasmacytoid dendritic cell (PDC) leukemia that can be loaded with relevant tumor antigen-derived peptides [8,9]. PDCs are a subtype of DCs playing an essential role in the immune response against viruses through type I interferon secretion [10,11,12]. They also have strong antigen presentation capabilities but have rarely been exploited as a cancer vaccine due to their scarcity [12,13,14].

In the context of melanoma, we showed that PDC*line cells loaded with peptides derived from four melanoma-associated antigens activated the rare tumor ASTCs present in the blood and, more importantly, in the tumors of patients [15,16]. These ASTCs proliferated, switched from a naïve to a memory phenotype, displayed cytotoxic potential, and killed autologous melanoma cells [16]. Vaccination experiments in humanized mice also demonstrated the immunostimulatory and tumoricidal potential of PDC*line-based vaccines ([15] and unpublished results). In 2020, we reported a first-in-human phase I/II study with melanoma patients (clinical trial number NCT01863108) treated with PDC*line cells loaded with four melanoma antigen-derived peptides [17]. The product, named “GeniusVac-Mel4”, was safe and well tolerated with encouraging signs of clinical activity [17]. Strikingly, a significant increase in the frequency of circulating memory ASTCs was observed, thereby demonstrating the priming and expansion of ASTCs by the GeniusVac-Mel4 vaccine in humans.

Given these promising results in melanoma, we endeavored to develop a new PDC*line-cell-based vaccine targeting NSCLC, which represents 80–85% of lung cancer cases. In this study, the capacity of PDC*line cells to trigger lung-tumor-specific CD8+ T-cell responses from patients’ PBMCs was assessed.

## 2. Results

### 2.1. PDC*Line Cells Induce a Broad Antitumor Response in Lung Cancer Patients

First, we selected shared tumor peptides, which were restricted to the HLA-A2 molecule and potentially overexpressed in lung cancer patients, using the Cancer Antigenic Peptide (CAPEP) Database (https://caped.icp.ucl.ac.be/; accessed on 28 January 2015) developed by the group of P. van der Brüggen [18]. Table 1 describes the 14 selected peptides. Although Wilms’ tumor antigen-1 (WT-1) was not a hit from the CAPEP database, it was selected based on the literature [19] and on its use in previous clinical trials with therapeutic vaccines in NSCLC [20].

We then evaluated the percentage of mRNA overexpression of all selected tumor antigens from which the corresponding peptides were derived within lung cancer patients with an in silico approach. For this, we used The Cancer Genome Atlas (TCGA) database and the GSE30219 transcriptomic data series, as described in Materials and Methods. Antigen mRNA overexpression was calculated in the two main subtypes of NSCLC: adenocarcinoma (AC) and squamous cell carcinoma (SCC).

The results presented in Figure 1 show a higher mRNA overexpression of cancer-germline antigens (CGAs) in SCC than in AC. GLULD1 was mostly found in AC.

Interestingly, survivin was largely overexpressed in both SCC and AC with 99% and 85% of cases, respectively. MUC1 and HER2 were expressed in nearly 100% of cases in both NSCLC subtypes.

The expansion of antigen-specific T cells (ASTCs) following the in vitro stimulation of patients’ PBMCs with peptide-loaded PDC*line cells was then assessed for all tumor peptides. Except in a few cases, the baseline frequency of tumor-specific precursors could not be evaluated, due to the low amount of collected PBMCs and the very low expected frequencies of circulating ASTC precursors in lung cancer patients (~0.001% or below; [37,38]). As PDC*line cells are potent in the priming and expansion of ASTCs, only IL-2 was added to the coculture to sustain the CD8+ T lymphocyte proliferation, as previously described [15,16].

PBMCs from 26 lung cancer patients were stimulated for 21–28 days with PDC*line cells loaded separately with the 14 selected tumor peptides plus Melan-A. Melan-A was used as the positive control because the frequency of circulating Melan-A-specific CD8+ T cells is known to be high in healthy donors [39], as well as in patients (~0.03%, unpublished data). Interestingly, all patients responded to Melan-A stimulation, attesting that patients’ PBMCs were fully able to be stimulated by peptide-loaded PDC*line cells. Regarding lung tumor peptides, as illustrated in Appendix A, the proportion of ASTCs in PBMCs was often below the limit of detection before coculture, as expected.

By contrast, after PDC*line stimulation, significant expansions of CD8+ T cells specific to all targeted antigens but MAGE-A1 (i.e., 13 out of 14 antigens) were clearly detected in 85% of patients (Figure 2 and Figure 3A, Table 2).

At the end of culture, the ASTC frequencies were high and variable depending on the targeted peptide (Figure 2). For example, the frequency of MAGE-A2- and MAGE-A9-specific T cells ranged from 0.026% to 0.128%, whereas that of MAGE-A3.2 ranged from 0.058% to 3.82%, which was the highest ASTC frequency obtained with tumor peptides.

As illustrated in Table 2 and Figure 3A, remarkably, 85% of patients (22/26) were able to mount T-cell responses toward at least one tumor antigen. Importantly, 69% of patients responded to at least two antigens and 39% to four or more antigens (Figure 3A). Expanded MAGE-A3.1- and WT-1-specific T cells were observed in only one patient, whereas survivin-, MAGE-A2-, and MAGE-A9-specific T-cell expansions were found in 58%, 64%, and 64% of patients, respectively (Table 2).

Interestingly, 29% to 43% of patients presented antitumor responses to MAGE-A3.2, MAGE-A4, MAGE-A10, NY-ESO-I, CAMEL, and GLULD1. Finally, expanded MUC1- and HER2-specific T cells were found in 11.5% and 15% of patients, respectively.

Taken together, these results demonstrate the strong ability of peptide-loaded PDC*line cells to induce expansion of tumor-specific T cells from lung cancer patients’ PBMCs.

### 2.2. PDC*Line Cells Induce Antitumor Response in Both Subtypes of NSCLC

We then wondered whether the observed antitumor antigen immune response depended on the subtype of NSCLC since among the 26 patients studied, 65% were adenocarcinoma (AC) and 23% were squamous cell carcinoma (SCC).

On average, each patient responded to 2.6 antigens, and there was no difference between the two groups of patients (Figure 3B).

When the responses were analyzed in detail, except for NY-ESO-1 and CAMEL antigens, the profiles of the induced immune response against each peptide were quite similar in the two groups (Figure 3C). Of note, there was no relationship identified between the response toward an antigen and the mRNA overexpression of this antigen in NSCLCs (Figure 1).

For example, although mRNAs encoding MAGE-A1, CAMEL, and NY-ESO-I were found to be overexpressed in a substantial proportion of SCC patients, no expansion of T cells specific to these tumor antigens was found in our study. Conversely, more than 50% of AC patients were able to develop antitumor responses against MAGE-A4, MAGE-A9, and CAMEL, although the mRNA overexpression (defined according to our threshold using TCGA database) of these antigens in AC was very low.

### 2.3. ICI Synergizes with Peptide-Loaded PDC*Line Cells to Expand Antitumor CD8+ T Cells

Since DC-based vaccines and ICIs have shown limited clinical benefits when used separately, the effect of the combination of both treatments is currently being measured in some clinical trials with NSCLC patients ([20] e.g., NCT04199559 and NCT03406715). We have, thus, investigated a possible synergistic effect of a clinical-grade antibody targeting PD-1 (Pembrolizumab) on the antigen-specific T-cell response induced by PDC*line cells. PDC*line cells were loaded with the peptides inducing the best T-cell responses (NY-ESO-1, CAMEL, MAGE-A2, MAGE-A3, and MAGE-A9) and used to stimulate PBMCs from eight new NSCLC patients. The cocultures were performed in the presence or absence of Pembrolizumab as the ICI. As shown in Figure 4, in the absence of the ICI, PDC*line cells successfully triggered the expansion of ASCTs specific to NY-ESO-I, MAGE-A3, and a mix of MAGE-A2/MAGE-A9/CAMEL in one, seven, and six patients, respectively. Strikingly, the addition of Pembrolizumab induced at least a doubling of the ASTC frequency, in five patients for one or several peptides (Figure 4B). As illustrated for patient #27, MAGE-A3-specific T cells increased threefold in the presence of the ICI (Figure 4A,B). Additionally, MAGE-A2/MAGE-A9/CAMEL-specific T cells more than tripled for patients #28, #29, and #31 when the antibody was added (Figure 4A,B). Interestingly, patients #34 and #31, who did not respond to PDC*line cells alone, presented a T-cell response specific to NY-ESO-1 and to the MAGE-A2/MAGE-A9/CAMEL mix, respectively, in the presence of the ICI.

These results demonstrate that the combination of ICI and PDC*line cells significantly improves the amplitude and the range of the specific T-cell response in lung cancer patients.

## 3. Discussion

Although immunotherapies, especially immune checkpoint inhibitors (ICIs), have brought impressive advances in the treatment of cancers, too many patients are still not eligible and/or do not respond to them. Therapeutic vaccines based on autologous dendritic cells induce good immune responses but are intrusive for patients and present limited reproducibility and clinical benefits (reviewed in [7]). Using allogeneic cell lines could overcome most of these problems. Indeed, the allogeneic plasmacytoid dendritic-cell-based cell line (PDC*line) we developed can induce a reproducible and efficient specific CD8+ cytotoxic T-cell response against tumors [15,16,17]. On the basis of robust and promising preclinical and clinical results in melanoma, we adapted the PDC*line-based antigen presentation platform to the most lethal cancer worldwide, i.e., lung cancer, particularly non-small-cell lung cancer (NSCLC). The goals of this study were to determine the efficiency of PDC*line cells to activate and expand circulating tumor-antigen-specific CD8+ T cells (ASTC) from NSCLC patients and to know if the PDC*line-based vaccine can synergize with ICIs to boost the antitumor response.

In this study, 14 different tumor-antigen-derived peptides were first selected based on their previous characterization and known immunogenicity. Interestingly, among them, MAGE-A9, GLULD-1, and CAMEL have never been tested in clinical trials with DC-based vaccines according to the “clinicaltrial.gov” website (accessed on 30 November 2022). Using peptide-loaded PDC*line cells, we observed the efficient activation end expansion of tumor ASTCs in 86% of patients, with 69% presenting two or more T-cell response specificities, which has never been observed before with PBMCs from NSCLC patients. The multi-specific T-cell response induced by PDC*line cells could be a great advantage in vivo to broaden and, thus, improve antitumor immune responses and prevent tumor escape.

Strikingly, despite the low frequency of specific precursors in PBMCs, PDC*line cells were able to greatly stimulate tumor ASTCs. With several antigens, the ASTC frequency was ten times higher than the limit of detection (0.01%), sometimes even exceeding 0.5% of CD8+ T cells, confirming the high potency of these professional antigen-presenting cells to stimulate antitumor immunity. As expected, and as shown by previous evidence in melanoma by Aspord et al., these expanded T cells are likely to be activated and functional, which unfortunately has not yet been tested in this study due to the lack of available cell material.

Until now, very few reports have described the expansion of antigen-specific T cells in lung cancer patients [40,41]. In a study by Groeper et al., the authors cultured tumor-infiltrating lymphocytes (TIL) from 33 stage I-III patients with autologous DCs and monitored the expansion of T cells specific to MAGE-A1, MAGE-A3, MAGE-A4, MAGE-A10, Multi-MAGE, and NY-ESO-1 [41]. Surprisingly, despite the use of TILs and the presence of cytotoxic activity in some samples, only MAGE-A10-specific T cells were detected in one patient after several weeks of culture. These results highlight the need to use potent antigen-presenting cells such as PDC*line cells to stimulate TILs, which may be more immunosuppressed than circulating T cells. Indeed, we have previously shown that PDC*line cells loaded with melanoma tumor peptides were able to generate an antitumor response using TILs with a high cytotoxic potential in a large number of melanoma patients [15,16]. In a study by Palata et al., antitumor responses were assessed with PBMCs from lung cancer patients directly stimulated with pools of MHC class I- and II-restricted peptides and revealed by intracellular IFN-γ staining [40]. Although it was not possible to precisely identify the peptides recognized by T cells, reactivity was found in most patients mainly against MAGE-A4.

In the present study, not all patients responded to all tumor peptides. For example, no or low T-cell expansions were observed for MAGE-A1, HER2, MAGE-A3.1, and WT-1. This could be explained by the paucity of precursors in the tested PBMCs and the low number of cells in coculture. Indeed, the frequency of tumor-specific precursors among CD8+ T cells is known to be below 0.001% [37,38], suggesting that no or very few precursors were put in coculture with PDC*line cells. Therefore, no expansion could take place in these cases. The lack of immunogenicity of some peptides and a strong tumor-induced immunosuppression of antigen-specific T cells could also explain these results.

In addition, we demonstrated that the mRNA overexpression of tumor antigens did not correlate with the percentage of patients who responded to the antigens of interest, which agrees with another study [40]. In line with this result, we showed that the ability to respond to PDC*line cell stimulation was not linked to the SCC or AC origin of cancer cells.

The ability of PDC*line cells to trigger a significant specific T-cell response in both subtypes of NSCLC is encouraging, especially for metastatic patients with SCC, who have worse prognoses than those with AC [42].

It has been strongly suggested that ICIs allow the unleashing of pre-existing immunity [4,5], and as a consequence, the combination of therapeutic vaccines with ICIs could be a powerful means by which to increase the clinical benefit of the two approaches. In our study, impressively, adding anti-PD-1 antibodies to cocultures of lung cancer patients’ PBMCs with peptide-loaded PDC*line cells greatly favored the expansion (two-to-nine-fold increase) of tumor-specific T cells with different specificities in five out of eight patients tested. Interestingly, although expansion was not always detectable in the presence of PDC*line cells alone, it was triggered in some cases when ICI was added to PDC*line cells. As mentioned above, this is in accordance with an immunosuppressed status of ASTCs that require the blocking of immune checkpoints to restore T-cell activation. These results are the first to show a synergistic effect of the combination of a potential therapeutic vaccine and anti-PD-1 antibodies on the T-cell response from patients with NSCLC ex vivo. In terms of functionality, the low number of specific T cells did not allow us to measure with certainty the expression of IFNγ and CD107a. However, the expanded T cells are expected to be fully functional. Indeed, as previously shown in in vitro and in vivo melanoma and virus models, peptide-loaded PDC*line cells induce the expansion of antigen-specific T cells with a strong cytotoxic potential as a bona fide professional antigen-presenting cell [15,16].

Despite intense research in the field, few DC-based therapeutic vaccines have demonstrated clinical benefits, in particular, in NSCLC. Using the PDC*line antigen presentation platform, we have shown that the circulating tumor-specific T cells of patients with NSCLC can be efficiently activated, against several tumor antigens simultaneously, and expanded. Importantly, the combination with anti-PD-1 antibodies further enhanced the activation potential of the platform. On the basis of this preclinical proof of concept, the PDC*line platform is currently being evaluated in combination with ICI in NSCLC (trial number NCT03970746).

## 4. Materials and Methods

### 4.1. Patients and Peripheral Blood Mononuclear Cells

Peripheral blood samples of patients (9 females and 25 males) with non-small-cell lung cancer (NSCLC) were collected between 2012 and 2016 at Grenoble Alpes University Hospital. The median age of the patients was 61.5 years. Blood samples were included in the biological sample collection DC-2011-1487 authorized by French health authorities. Thirty-four patients expressing the HLA-A*02:01 allele were included in the study. All patients signed informed consent. The study of the T-cell response induced by PDC*line cells was performed with the blood of 26 patients (patients 1–26), and the effect of immune checkpoint inhibitors was studied with the blood of 8 patients (patients 27–34). The clinical characteristics of the patients are summarized in Table 3.

Peripheral blood mononuclear cells were purified from the patients’ blood using Ficoll-Hypaque density gradient centrifugation (Lymphocyte Separation Medium, Eurobio) and stored frozen.

### 4.2. Preparation of Tumor-Peptide-Loaded PDC*Line

The origin of PDC*line cells and their culture in X-VIVO 15 medium has been previously described [8,9,16]. PDC*line cells were separately loaded with 14 HLA-A*02:01-restricted tumor peptides for 3 h, as previously described [15,16,38]. Briefly, PDC*line cells (1 million/mL) were incubated for 3 h at 37 °C with 10 µM of each peptide in X-VIVO-15. The tumor peptides were purchased from NeoMPS or JPT technologies. The characteristics of the peptides are detailed in Table 2 in the Results section. After loading, the cells were washed twice with X-VIVO 15, resuspended at 0.2 million/mL, mixed in equal ratios in pools of 6, and irradiated at 60 Gy. The first mix consisted of cells loaded with MUC-1-, HER2-, survivin-, WT-1-, and MAGE-A3.2-derived peptides. The second mix consisted of cells loaded with MAGE-A1-, MAGE-A2-, MUC-1-, GLULD-1-, MAGE-A4-, and NY-ESO-1-derived peptides. The third one consisted of cells loaded with HER2-, CAMEL-, MAGE-A3.1-, MAGE-A3.2-, MAGE-A9-, and MAGE-A10-derived peptides. Positive control with Melan-A-derived peptide (ELAGIGILTV) was performed either alone or included in the first mix. Depending on experiments, one or two mixes were then cocultured separately with patients’ PBMCs, at a ratio of 2 million PBMCs for 0.2 million pooled PDC*line cells per well.

For the study with anti-PD-1 antibodies, PDC*line cells were loaded individually with NY-ESO-1-, MAGE-A3.1-, MAGE-A3.2-, MAGE-A2-, MAGE-A9-, and CAMEL-derived peptides, then mixed and irradiated at 60 Gy.

### 4.3. Patients’ PBMC Stimulation with Loaded PDC*Line Cells

Two million PBMCs were cocultured with 0.24 × 10^6^ peptide-loaded PDC*line cells in 1 mL RPMI 1640 (Gibco, Life Technologies, France) supplemented with non-essential amino acids (Gibco, Life Technologies, France), 1 mM sodium pyruvate (Sigma, Saint-Quentin Fallavier, France), 20 µg/mL gentamycin, and 10% fetal calf serum (FCS, Gibco, Life Technologies, France). In all but 6 cultures, up to 200 IU/mL IL-2 (Peprotech, Neuilly-sur-seine, France) was added at the beginning of the cultures. All the cocultures were re-stimulated weekly with loaded PDC*line cells and IL-2 and collected after 21 or 28 days.

For the study with anti-PD-1 antibodies, one million PBMCs were cultured with 0.24 × 10^6^ peptide-loaded PDC*line cells in the presence of 200 IU/mL IL-2 (Proleukin, Novartis, Switzerland) for 14 days with a restimulation with loaded PDC*line cells on the seventh day of culture. Half of the cocultures were treated with 10 µg/mL anti-PD1 antibody (Pembrolizumab, ^®^Keytruda, Merck, France).

### 4.4. Detection of Antigen-Specific CD8+ T Cells

Patients’ PBMCs and the cells collected at the end of cocultures were resuspended in PBS (Gibco, Life Technologies, France) with 2% decomplemented FCS and incubated with fluorochrome-conjugated antigen-specific multimers for 20 min in the dark at room temperature. Either tetramers (ITag, Beckman Coulter, Villepinte) or dextramers (Immudex, Denmark) were used as multimers. For the study with anti-PD-1 antibodies, a mix of dextramers specific to MAGE-A2, MAGE-A9, and CAMEL conjugated to the same fluorochrome (allophycocyanin) was used to label CD8-positive T cells. In the same tube, a mix of dextramers specific to MAGE-A3.1 and MAGE-A3.2 conjugated to the same fluorochrome (phycoerythrin) was added, in addition to fluorescein isothiocyanate-conjugated NY-ESO-1-specific dextramers.

After washing, BV421-conjugated anti-CD3 (BD Biosciences, Le Pont de Claix, France) and PerCP-Cy5.5-conjugated anti-CD8 antibodies (BD Biosciences, Le Pont de Claix, France) were added and incubated for 20 min in the dark at 4 °C. Viability dye (Live and Dead, Fisher Scientific, Illkrich, France) was added to the antibody mix. The cells were washed and resuspended in FACS Lysing solution (BD Biosciences, #349202) before fluorescence acquisition with a flow cytometer (BD FACS Canto II) and analysis with FlowJo software (Tree Star, Inc., Ashland, OR, USA). The frequency of multimer-positive cells was measured in the CD8-positive living single-cell population of lymphocytes, as described in Appendix A. Owing to the low number of cells collected after expansion, the limit of detection of ASTCs was set at 0.01% of total gated CD8+ T cells.

### 4.5. Transcriptomic Data Analysis

RNA expression patterns of the genes in normal lung and in lung cancer samples were analyzed in the GSE30219 transcriptomic series obtained by the Affymetrix Human Genome U133 Plus 2.0 Array and in The Cancer Genome Atlas (TCGA) series, TCGA-LUAD, and TCGA-LUSC, obtained by RNA-seq technology. The GSE30219 series (https://www.ncbi.nlm.nih.gov/geo/query/acc.cgi?acc=GSE30219; accessed on 30 July 2020) includes 14 normal lung tissues, 86 adenocarcinomas (ACs), and 60 squamous cell carcinoma (SCC) samples. The TCGA data set includes 87 normal lung samples, 433 AC, and 430 SCC samples (https://www.cancer.gov/about-nci/organization/ccg/research/structural-genomics/tcga; accessed on 30 July 2020). In addition, we used the data of 48 different normal tissues provided by the GTEX portal and NCBI Sequence Read Archive in RNA-seq technology (datasets PRJNA280600, PRJEB4337, PRJEB2445, PRJNA270632, GSE70741, GSE53096). The data of the microarray dataset GSE30219 were normalized using Robust Multi-array Average (RMA) and then log-transformed in log base 2 scale. RNA-seq data were normalized by the FPKM method and then log-transformed by taking log2(1 + FPKM).

All the genes used in this study except HER2 and MUC1 normally have a tissue-predominant expression profile in testis, placenta, and embryonic stem cells and are not, or are lowly, expressed in normal lungs. In cancers, these tissue-specific genes can be ectopically activated due to epigenetic deregulations during the oncogenesis [43]. We analyzed the transcriptomic data of these genes according to the strategy described by Rousseaux et al. [43]. For each gene, the percentage of tumor samples was calculated with the gene mRNA expression over a threshold defined according to the distribution of values in normal lungs. In order to measure the proportion of overexpression in tumor samples, the threshold was placed at the mean of expression in normal samples plus 2 standard deviations.

For HER2 and MUC1 genes, whose mRNAs are ubiquitously expressed in different normal tissues, the objective was to define whether the genes were expressed or not in tumor samples. For this purpose, we used a fixed threshold of expression calibrated in the dataset of normal tissues GTEX and corresponding to the signal level in reference (fetal) tissues in which the two genes were not expressed. The percentage of samples expressed above the threshold was then calculated in tumor samples.

## Figures and Tables

**Figure 1 ijms-24-01897-f001:**
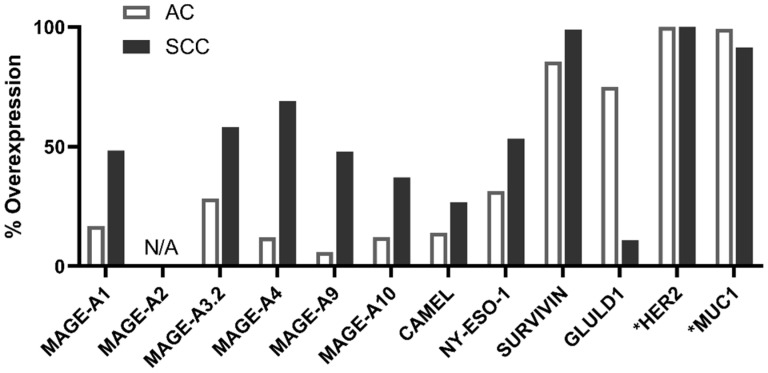
mRNA overexpression frequency of selected antigens in NSCLC subtypes. Gene overexpression frequencies in tumoral versus non-tumoral lung samples were calculated after the analysis of the TCGA data set, except for GLULD1 and NY-ESO-1; the frequencies of which were calculated using the GSE30219 data series. * For HER2 and MUC1 genes, the percentage of expression—and not overexpression—frequency is shown because their mRNAs are ubiquitously expressed in normal tissues. A threshold corresponding to the signal in fetal tissues which do not express the two genes was used for the calculation. N/A—not applicable. AC—adenocarcinoma. SCC—squamous cell carcinoma.

**Figure 2 ijms-24-01897-f002:**
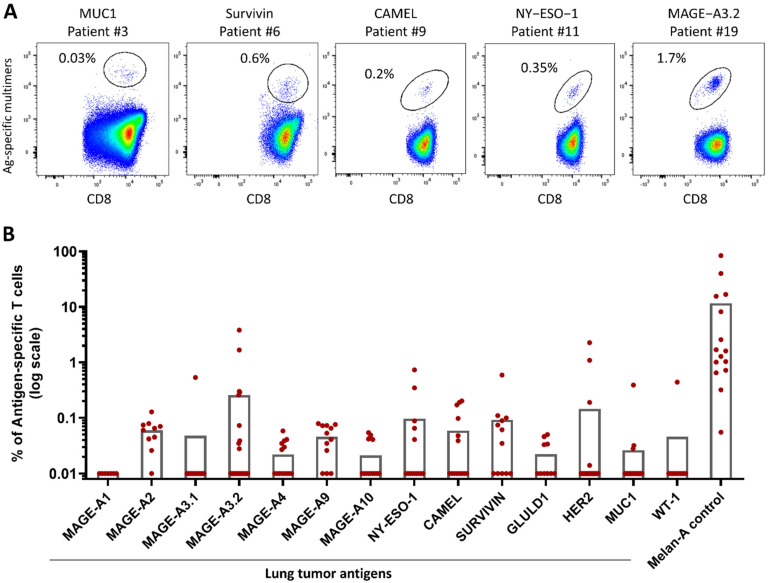
Specific CD8+ T-cell responses induced by PDC*line cells in lung cancer patients. PDC*line cells were loaded individually with selected HLA-A*02-restricted tumor peptides, mixed, and irradiated. The cells were cultured with patients’ PBMCs, and the proportion of antigen-specific CD8-positive T cells (ASTC) was measured using fluorescent multimers specific to each HLA-A*02-restricted peptide. (**A**) Examples of multimer labeling of ASTCs after coculture of patients’ PBMCs with loaded-PDC*line cells. Positive responses of five different patients are shown for five different antigens with pseudocolor dot plots made with FlowJo software. (**B**) Frequencies of ASTCs for all antigens after culture of patients’ PBMCs with tumor-peptide-loaded PDC*line cells. Each red dot represents a patient. When ASTCs were not detected, the frequency was the limit of detection, i.e., 0.01%. The bars show the mean of the frequencies.

**Figure 3 ijms-24-01897-f003:**
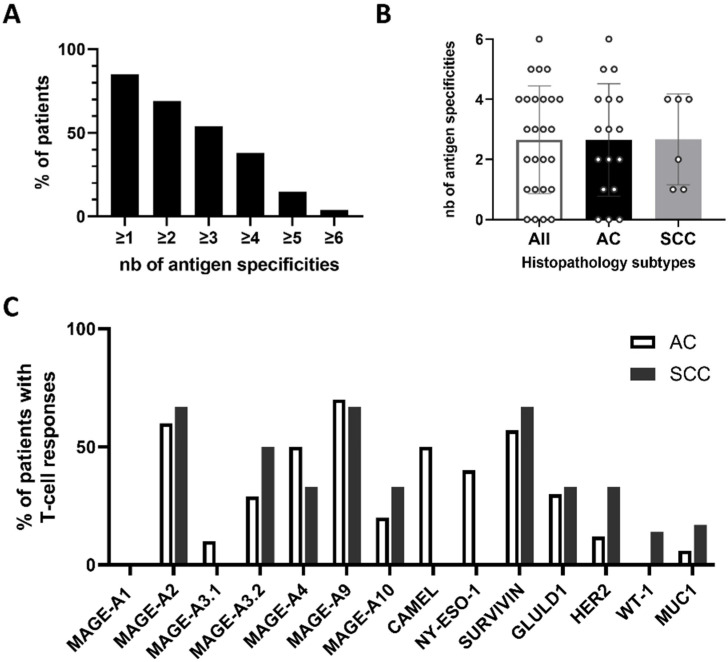
Specific T-cell responses according to NSCLC subtypes. (**A**) Proportion of patients responding to more than one antigen. (**B**) Number of antigen-specific T-cell responses per patient according to their histopathology subtype. Each circle represents a patient. The mean is shown as +/− standard deviation. AC—adenocarcinoma; SCC—small cell carcinoma. (**C**) Proportion of patients with specific T-cell responses in the two groups of patients.

**Figure 4 ijms-24-01897-f004:**
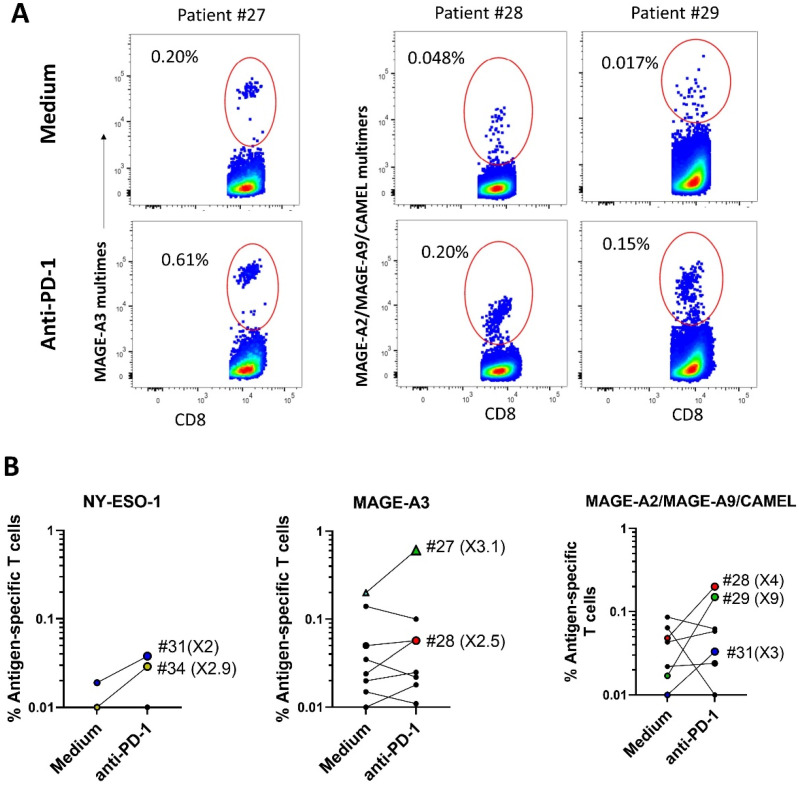
Effect of anti-PD-1 antibodies on the PDC*line-cell-induced T-cell response in NSCLC patients. Patients’ PBMCs and tumor-peptide-loaded PDC*line cells were cultured in the presence or absence of anti-PD-1 antibodies. The proportion of antigen-specific CD8+ T cells (ASTC) was measured by fluorescent multimer labeling at the end of the coculture. To detect MAGE-A2-, MAGE-A9- and CAMEL-specific ASTCs, a mix of specific dextramers conjugated to the same fluorochrome was used. (**A**) Illustrative pseudocolor dot plots showing the effect of anti-PD-1 antibodies on the proportion of ASTCs for three different patients. (**B**) Effect of anti-PD-1 antibodies on the expansion of specific ASTCs in the whole population of patients. Colored symbols indicate a positive effect of anti-PD-1 antibodies, and each color represents a different patient (e.g., patient #31 is represented with a blue circle). Increases greater than twofold are indicated in brackets.

**Table 1 ijms-24-01897-t001:** List of the selected lung tumor antigens.

Antigen Type	Antigen	Peptide Sequence (HLA-A*02:01)	References
Cancer-germline Antigens	MAGE-A1	_278_KVLEYVIKV_286_	[21,22]
	MAGE-A2	_157_YLQLVFGIEV_166_	[23]
	MAGE-A3.1	_112_KVAELVHFL_120_	[23]
	MAGE-A3.2	_271_FLWGPRALV_279_	[24]
	MAGE-A4	_230_GVYDGREHTV_239_	[25]
	MAGE-A9	_223_ALSVMGVYV_231_	[26]
	MAGE-A10	_254_GLYDGMEHL_262_	[27]
	NY-ESO-1 (CTAG1B)	_157_SLLMWITQC_165_	[28,29,30]
	CAMEL (CTAG2)	_152_MLMAQEALAFL_162_	[31]
Overexpressed Antigens	GLULD1 (LGSN)	_270_FLPEFGISSA_279_	[32]
	SURVIVIN (BIRC5)	_95_ELTLGEFLKL_104_	[33,34]
	HER2 (ERBB2)	_369_KIFGSLAFL_377_	[35]
	WT-1	_126_RMFPNAPYL_134_	[19]
Post-translational	MUC-1	_12_LLLLTVLTV_20_	[36]

Abbreviations: MAGE-A—melanoma-associated antigen-A; NY-ESO-1—New York esophageal cancer; CTAG—cancer–testis antigen; CAMEL—cytotoxic T-lymphocyte-recognized antigen on melanoma; GLULD-1—glutamate–ammonia ligase domain-containing protein 1; HER2—human epidermal growth factor receptor-2; LGSN—Lengsin; BIRC5—baculoviral IAP repeat containing 5; ERBB2—erb-B2 receptor tyrosine kinase 2; WT—Wilms’ tumor; and MUC—mucin.

**Table 2 ijms-24-01897-t002:** Proportions of patients responding to tumor antigens (Ags).

MAGE-A1	MAGE-A2	MAGE-A3.1	MAGE-A3.2	MAGE-A4	MAGE-A9	MAGE-A10	NY-ESO-1	CAMEL	GLULD1	SURV	HER2	MUC-1	WT-1	At Least One Ag
0/14	9/14	1/14	9/26	6/14	9/14	4/14	5/14	6/14	4/14	7/12	4/26	3/26	1/12	22/26
0%	64%	7.1%	34.6%	43%	64%	29%	36%	43%	29%	58%	15%	11.5%	8.3%	84.6%

**Table 3 ijms-24-01897-t003:** Clinical characteristics of the patients included in the study.

Patient Number	Histology	Stage	Smoking Status
1	SCC	IIB	Active
2	AC	IIIA	Active
3	SCC	IA	Active
4	AC	IA	Former
5	AC	IIA	Non-smoker
6	TC	IB	Former
7	SCC	IIA	Active
8	SCC	IB	Active
9	AC	IIIA	Active
10	AC	IIB	Former
11	AC	IIIA	Active
12	AC	IIIA	Active
13	AC	IA	Active
14	AC	IV	Active
15	LCNC	IIIA	Active
16	AC	IIB	Active
17	AC	IA	Active
18	AC	IB	Active
19	SCC	IIB	Active
20	AC	IB	Former
21	AC	IIA	Active
22	AC	IA	Active
23	TC	IA	Active
24	AC	nd	Former
25	SCC	nd	Unknown
26	AC	II	Former
27	SCC	IIB	Active
28	SCC	IIIA	Active
29	AC	IIb	Former
30	AC	IIB	Former
31	AC	IIA	Former
32	AC	IB	Former
33	AC	IIB	Non-smoker
34	SCC	IV	Former

Abbreviations: AC—adenocarcinoma; SCC—small cell carcinoma; LCNC—large cell neuroendocrine carcinoma; TC—typical carcinoid; and nd—not determined.

## Data Availability

The Cancer Genome Atlas (TCGA) is publicly available: https://www.cancer.gov/about-nci/organization/ccg/research/structural-genomics/tcga; accessed on 30 July 2020.

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
