# Peer review of "A New Plasmacytoid Dendritic Cell-Based Vaccine in Combination with Anti-PD-1 Expands the Tumor-Specific CD8+ T Cells of Lung Cancer Patients"

_ijms, 2023, doi:10.3390/ijms24031897_

Round 1
Reviewer 1 Report
Overall Comments:
The goal of manuscript entitled, “A New Plasmacytoid Dendritic Cell-Based Vaccine in Combination with Anti-PD1 Expands the Tumor-Specific CD8+ T Cells of Lung Cancer Patients” by Hannani et. al. was to demonstrate the potential of this platform to activate circulating tumor-specific CD8+ T cells in patients with lung cancer, specifically non-small-cell lung cancer (NSCLC). This goal was partially achieved. Although the authors provide evidence of tumor antigen specific CD8+ T cell expansion, the activation and functional statuses of those T cells remains undefined. Further, a key control condition, the PDC*line in the absence of antigen loading needs to be included. If the authors can provide compelling evidence supporting the use of the antigen-loaded PDC*line to increase antigen-specific CD8+ T cell activation, expansion, and effector function, this reviewer will reconsider the manuscript for publication in International Journal of Molecular Sciences.
Specific Comments:
1.) Minor – please change an-ti-PD1 in the abstract to ‘anti-PD1’
2.) How is known that the lung cancer antigens are loaded and presented by the PDC*line? Could the authors please provide these results; PDC*line expressing lung tumor specific antigens described?
3.) Similar to 2.) what is the activation status of PDC*line after antigen treatment? i.e. levels of CD40, CD69, CD80, CD86? Could the authors provide these results?
4.) Fig. 2 – although the ag-multimers suggest antigen-specific T cell expansion, a proliferation experiment should be conducted where the CD8+ T cell patient compartment alone and in combination with PBMCs are mixed with the ag-pulsed PDC*line to determine how well the CD8+ ag-specific T CD8+ T cell are expanding in response to the PDC*line expressing tumor antigens.
5.) Although antigen specific CD8+ T cell expansion data is provided, the activation status remains unclear. What are the levels of IL-2 and IFN-gamma produced? What are surface expression levels of CD25 and CD69 post co-culture with the antigen pulsed PDC*line and CD8+ T cells (PBMCs)? A detailed evaluation of the T cell activation status should be provided.
6.) Fig. 3B – the reviewer is unclear what is meant by ‘Pathology’ is this referring to the exact phenotype of the lung cell cancer? Adenocarcinoma vs. Squamous cell carcinoma? Perhaps using the work ‘Phenotype’ may be clearer.
7.) The sentence on page 6 of the manuscript, “We then wondered whether the observed antitumor response depended on the sub- 181 type of NSCLC, since among the 26 patients studied, 65% were adenocarcinoma (AC) and 182 23% were squamous cell carcinoma (SCC).”
should be revised to, “We then wondered whether the observed anti-tumor antigen immune response depended on the sub- 181 type of NSCLC, since among the 26 patients studied, 65% were adenocarcinoma (AC) and 182 23% were squamous cell carcinoma (SCC).”
otherwise, readers would think that the authors measured an anti-tumor killing response.
8.) Although the authors provide evidence of tumor-antigen specific T cell expansion, whether the T cells are functional remains unknown. A set of experiments looking at the ability of CD8+ antigen specific T cells to kill antigen-pulsed target cells should be evaluated.
9.) Fig. 4- the ability to influence the % of antigen-specific T cell expansion by utilizing anti-PD1 is interesting. How does the increase in % correlate to the number of cells expanded? Similar to comment 4.) a proliferation experiment labeling CD8+ T cells with proliferation dye could be useful to monitor the proliferation status of the antigen-specific T cells and test the impact of anti-PD1 blockade. Also, how does the presence of anti-PD1 influence the activation status (CD69, CD25 surface expression; IL-2, IFN-gamma secretion)? These results should be provided.
10.) Antigen-specific T cell expansion seems to be evaluated at single time points. To better understand the timing of expansion and activation, the authors should execute experiments looking at antigen specific CD8+ T cell expansion and activation status as a function of time and include at least 3 time points. These results should be provided.
11.) Figs. 2-4: throughout, a key control condition is missing: the PDC*line in the absence of antigen needs to be included.
Author Response
We would like to thank the reviewers for their suggestions and comments. We did our best to clarify the topic and answer to the reviewer’s questions.

Reviewer 2 Report
In this manuscript, Hannani et al. aim to develop a new method of immunization with tumor antigens, in order to induce antigen specific CD8+ T cell responses from patient samples. For this, the authors use a plasmacytoid DC-based immunization method. Results are clearly presented and the conclusions presented match the results shown. This is an interesting work that, however, could benefit from additional assessments since at this point it is not clear if these CD8+ T cells are indeed functional.
Specific points to be addressed:
1. A thorough assessment of the cytokines produced by antigen specific CD8+ T cells would be important. Is this method inducing polyfunctional CD8+ T cells? I.e., cells producing IFNg, GzmB, TNFa and/or IL2?
2. Likewise, it is not clear if the cells elicited by this immunization display a terminal effector, memory-like or exhausted phenotype. One would assume a mix of effector and memory-phenotype cells are generated, but assessments of CD45 isoforms, as well as CCR7, CD62L etc. would be welcome.
3. Finally, a functional assessment of these cells after co-culture with tumor lines would be a nice proof-of-concept that these cells are indeed able to perform efficient antitumor functions.
Author Response

(The authors gave the same response as above.)

Round 2
Reviewer 1 Report
Hannani et. al. present a revised manuscript demonstrating the potential of the PDC*DC cell line platform to activate circulating tumor-specific CD8+ T cells in patients with lung cancer, specifically non-small-cell lung cancer (NSCLC). The authors provide evidence of tumor antigen specific CD8+ T cell expansion, and describe previous evidence of activation and functionality. The authors satisfactorily addressed this reviewer's comments in the author's response, in particular the limited availability of clinical sample material for experimentation. Once the authors clearly state in their discussion that their expanded T cells are likely activated and functional due to previously defined evidence, yet not tested due to lack of available material, their work will be suitable for publication in International Journal of Molecular Sciences.
Author Response
We thank the reviewer for his understanding and consideration of our responses.
As requested, this following sentence has been added in the fourth paragraph of the discussion:
"As expected, and as shown by previous evidence in melanoma by Aspord et al, these expanded T cells are likely to be activated and functional, which unfortunately has not yet been tested in this study due to the lack of available cell material."
Reviewer 2 Report
The authors addressed all my comments properly. I recommend acceptance of this manuscript.
Author Response
We thank the reviewer for his understanding and consideration of our responses.
We thank the reviewer for the final acceptance of the manuscript